

# Comparison of therapeutic effects of different mesenchymal stem cells on rheumatoid arthritis in mice

Qing Zhang[1,2,3,*], Qihong Li[4,*], Jun Zhu[2,3], Hao Guo[2,3], Qiming Zhai[1,2,3], Bei Li[2,3], Yan Jin[2,3], Xiaoning He[2,3] and Fang Jin[1,2,3]

[1] State Key Laboratory of Military Stomatology & National Clinical Research Center for Oral Diseases & Shaanxi Clinical Research Center for Oral Diseases, Department of Orthodontics, School of Stomatology, The Fourth Military Medical University, Xi'an, Shaanxi, China
[2] State Key Laboratory of Military Stomatology & National Clinical Research Center for Oral Diseases & Shaanxi International Joint Research Center for Oral Diseases, Center for Tissue Engineering, School of Stomatology, The Fourth Military Medical Universit, Xi'an, Shaanxi, China
[3] Xi'an Institute of Tissue Engineering and Regenerative Medicine, Xi'an, Shaanxi, China
[4] Department of Stomatology, the Fifth Medical Centre, Chinese PLA General Hospital, Beijing, China
* These authors contributed equally to this work.

Corresponding authors
Xiaoning He, hxn_800222@163.com
Fang Jin, jinfang@fmmu.edu.cn

## ABSTRACT

**Background:** Rheumatoid arthritis (RA) is a chronic and nonspecific autoimmune disease, which leads to joint destruction and deformity. To investigate the potential of human mesenchymal stem cells (MSCs) as a new therapeutic strategy for patients with RA, we compared the therapeutic effects of bone marrow derived MSCs (BMSCs), umbilical cord derived MSCs (UCs), and stem cells derived from human exfoliated deciduous teeth (SHED) on collagen-induced arthritis (CIA) in mice.
**Methods:** A total of 24 DBA/1 mice were infused with type II collagen to induce RA in the experimental model. MSC-treated mice were infused with UCs, BMSCs, and SHED, respectively. Bone erosion and joint destruction were measured by micro-computed tomographic (micro-CT) analysis and hematoxylin and eosin staining. The levels of tumor necrosis factor α (TNF-α) and interleukin-1β (IL-1β) were measured by immunohistochemistry and Enzyme-Linked Immunosorbent Assay (ELISA).
**Results:** Systemic delivery of MSCs significantly improved the severity of the symptoms related to CIA to greater extent compared with the untreated control group. Micro-CT revealed reduced bone erosions in the metatarsophalangeal joints upon treatment with MSCs. Additionally, according to histologic evaluation, reduced synovitis and articular destruction were observed in MSC-treated groups. The levels of TNF-α and IL-1β in the serum and joints decreased with treatment by MSCs.
**Conclusion:** Our findings suggest that systemic infusion of UCs, BMSCs, and SHED may significantly alleviate the effects of RA. The therapeutic effect of BMSCs was greater than that of SHED, while the UCs were shown to have the best therapeutic effect on CIA mice. In conclusion, compared with BMSCs and SHED, UCs may be a more suitable source of MSCs for the treatment of patients with RA.

## INTRODUCTION

Rheumatoid arthritis (RA), a chronic and nonspecific inflammation, is characterized by recurrent, progressive joint pain, and swelling. It attacks joints throughout the entire body, which may lead to joint destruction and eventual deformity (*Scott, Wolfe & Huizinga, 2010*). Bone loss at the articular end, joint space stenosis, and worm-like erosions on the articular or bone margin, are the typical X-ray features of RA (*Arnett et al., 1988*). According to statistics from the World Health Organization, the global incidence rate of RA is 0.5–1% of the population and the disability rate over the course of 15 years is 61.3% (*Smolen, Aletaha & McInnes, 2016*). RA not only causes a decline in patients' physical function, quality of life, and social participation, but also brings a huge economic burden to the families of these patients and society at large.

Currently, it is widely accepted that immune dysfunction plays a significant role in the pathogenesis and course of RA (*Abramson & Amin, 2002*). Characteristic of RA, tumor necrosis factor $\alpha$ (TNF-$\alpha$) causes acute and chronic inflammation of the synovium and ultimately leads to tissue destruction. TNF-$\alpha$ also induces the release of other cytokines, including interleukin-1 (IL-1), IL-6, and chemokines. TNF-$\alpha$ and IL-1 not only changes the function of synovial fibroblasts to cause them to secrete a variety of inflammatory mediators such as matrix metalloproteinase IL-6 and prostaglandin E, but also enables synovial macrophages to differentiate into osteoclasts and help activate osteoclasts to absorb bone (*Fox, 2000*; *Lam et al., 2000*). Many nonsteroidal anti-inflammatory drugs, disease modifying antirheumatic drugs (DMARDs), and glucocorticoids have been used to treat RA. However, these drugs do not work on all patients. The drugs mentioned above present some shortcomings related to the incidence of side effects and high recurrence rates of RA (*Joensuu et al., 2015*). Therefore, it is important to develop a new and more effective therapy for RA.

For the last several years, mesenchymal stem cells (MSCs) have been widely studied and implemented as a new therapeutic tool for an increasing number of clinical diseases. MSC with strong immunomodulatory and anti-inflammatory effects can repair damaged tissue by regulating the local environment through cellular interactions and via the secretion of multiple factors. These characteristics make MSC an ideal tool for the treatment of autoimmune diseases (ADs), such as RA (*Maumus, Jorgensen & Noël, 2013*; *Kastrinaki & Papadaki, 2009*). The therapeutic effect of MSC on RA is attributed to the regulation on immune cells and inflammatory cytokines involved in the course of RA. MSC inhibit the proliferation and activation of T-lymphocyte and B-cell by secretion of cytokines (paracrine effect) and cell–cell direct contact effect (*Di Nicola et al., 2002*; *Soleymaninejadian, Pramanik & Samadian, 2012*; *Mohammadzadeh et al., 2014*; *Franquesa et al., 2015*; *Corcione et al., 2006*; *Schena et al., 2010*). Meanwhile, MSC could also transform M1-type pro-inflammatory macrophage into M2-type anti-inflammatory macrophage. Thus, the expression of pro-inflammatory cytokines was reduced while the expression of pro-inflammatory cytokines was increased (*Németh et al., 2009*). According to previous studies, MSC infusion through venous or intraperitoneal injection have shown positive effects on animals and human patients with RA

(*Swart & Wulffraat, 2014*). Although clinical trials on MSC therapy in RA are limited, the safety and effectiveness of this treatment measures in RA have been proved. Intravenous infusions of allogeneic adipose-derived MSCs in 46 patients with active refractory RA were in general well tolerated without evidence of toxicity over 3 months (*Álvaro-Gracia et al., 2017*). The treatment of intravenous infusion of allogeneic bone marrow derived MSCs (BMSCs) or umbilical cord derived MSCs (UCs) into four RA patients who were resistant to DMARDs was safe and resulted in partial clinical improvement (*Liang et al., 2012*). Intravenous injection of UCs in addition to DMARDs induced a significant clinical improvement in patients who had inadequate responses to traditional medication (*Wang et al., 2013*). Although most animal studies and limited clinical trials proved the positive effects of MSC in RA, few studies indicated that MSC did not confer any benefit in RA (*Schurgers et al., 2010*) or even aggravated RA (*Chen et al., 2010*), particularly when the cells were applied at a point where inflammation is high (*Djouad et al., 2005*).

BMSCs UCs, and stem cells derived from human exfoliated deciduous teeth (SHED) are the three most widely used MSC for treatment at present. All three MSC populations above are able to be productively and safely isolated and purified. In the past, BMSCs were considered the most common cell source for stem-cell-based therapy. However, aspirating bone marrow is an invasive and painful procedure. In addition, there are some limitations in using BMSCs due to the high degree of viral exposure and the significant decrease in cell number and proliferation/differentiation capacity with the increasing age of the donor (*Stolzing et al., 2008*; *Rao & Mattson, 2001*; *Wakitani, Saito & Caplan, 1995*; *Young et al., 1999*). On the contrary, UCs and SHED, derived from discarded human tissues are able to be noninvasively isolated from umbilical cords and exfoliated deciduous teeth, respectively. Compared to BMSCs, UCs and SHED present with better proliferation ability and greater potential for differentiation were considered better cell sources for therapeutic applications (*Al-toub et al., 2013*; *Hunt, 2011*; *Nakamura et al., 2009*). Previous animal studies and clinical trials reported BMSCs and UCs could have a curative effect on RA (*Liu et al., 2010*). But until now, there were no reports on the curative effect of SHED on RA. There have also been no comparative analyses on the therapeutic efficacy of these three MSC populations on RA.

In this paper, we not only explored the curative effects of SHED on RA, but also compared the efficacy of BMSCs, UCs, and SHED on RA, expecting to provide a basis for the use of different MSCs in the clinical treatment of RA or other ADs.

## MATERIALS AND METHODS

### Animals

DBA/1 mice were purchased from Beijing Vital River Laboratory in Beijing, China. All mice were 8-weeks-old and were housed four to five per cage under specific pathogen-free conditions (22 °C, 50–55% humidity, and 12 h light/12 h dark cycles) with ready access to food and water. All animal-handling procedures and experimental protocols performed were approved by the guidelines set forth by the Animal Care Committee of the Fourth Military Medical University, Xi'an, China (2018-kq-007).

## Isolation and culture of MSCs

The collection of human tissues was approved by the Ethics Committee of the Fourth Military Medical University (IRB-REV-2013-002). The umbilical cords were cut into segments (two to three cm) after rinsed several times in sterile phosphate-buffered saline (PBS). Cord vessels, including two arteries and one vein, were removed. Then the umbilical cords were cut into small pieces (0.5–1 cm$^3$) and placed directly into 10 cm culture dishes for culture expansion in basal culture medium. After 5 days, non-adherent cells were discarded and adherent cells were continued to culture (*Mitchell et al., 2003*; *Qiu et al., 2018*; *Shang et al., 2017*). Dental pulps from human exfoliated deciduous teeth were cut into small pieces and digested with collagenase I (Gibco, Grand Island, NY, USA) for 1 h at 37 °C. Single-cell suspensions were cultured in basal culture medium (*Miura et al., 2003*; *Xuan et al., 2018*). Cells were purified from the bone marrow aspirates of the iliac crest using the Percoll density gradient centrifugation method and cultured in basal culture medium (*Liao et al., 2017*; *Liu et al., 2018*). The basal culture medium was composed of α-MEM medium (Gibco, Grand Island, NY, USA), 10% FBS (Sijiqing, Hangzhou, China), 2 mM L-glutamine (Invitrogen, Carlsbad, CA, USA), 100 U/mL penicillin, and 100 U/mL streptomycin (Invitrogen, Carlsbad, CA, USA). Single-cell suspensions were equally seeded in 10 cm dishes (Thermo, Suzhou, China) and initially maintained in an atmosphere of 5% $CO_2$ at 37 °C. The medium was changed every 3 days until the adherent cells were 80–90% confluent. Then MSCs were passaged after digestion with 0.25% trypsin. MSCs at passage three were used for the tests in this research.

## Evaluation of surface markers by flow cytometry

Eight cell surface markers (CD105, CD73, CD90, CD45, CD34, CD14, CD19, and HLA-DR) were evaluated to identify MSCs according to ISCT criteria (*Dominici et al., 2006*). MSCs were counted to ensure each cell suspension had more than $1 \times 10^6$ cells. The cells were incubated with human anti-CD105 (PE), anti-CD73 (FITC), anti-CD90 (PE), anti-CD45 (PE), anti-CD34 (PE), anti-CD14 (FITC), anti-CD19 (APC), and anti-HLA-DR (FITC) (all from eBioscience, San Diego, CA, USA), respectively for 30 min at room temperature. The cells were then washed and suspended for flow cytometry analysis.

## Osteogenic differentiation assay

Mesenchymal stem cells were seeded in six-well plates and cultured in basal medium for 24 h. Then, cells were cultured for 28 days with osteogenic medium, containing 10 mM β-glycerophosphate (Sigma-Aldrich, St. Louis, MO, USA), 50 μg/ml ascorbic acid (Sigma-Aldrich, St. Louis, MO, USA) and 100 nM dexamethasone (Sigma-Aldrich, St. Louis, MO, USA). The medium was refreshed every 3 days. The cells washed twice with PBS to remove medium and then fixed with 4% paraformaldehyde (Sigma-Aldrich, St. Louis, MO, USA). Then cells were washed gently three times with ddH2O and stained with 1% Alizarin Red (Sigma-Aldrich, St. Louis, MO, USA). Photographs were taken by an inverted microscope (Olympus, Tokyo, Japan).

## Adipogenic differentiation assay

Mesenchymal stem cells were seeded in six-well plates and cultured in basal medium for 24 h. Then, cells were cultured for 14 days with adipogenic medium, containing 0.5 mM isobutylmethylxanthine (MP Biomedicals, Santa Ana, CA, USA), 10 μg/ml insulin, 0.5 mM dexamethasone (MP Biomedicals, Santa Ana, CA, USA) and 60 mM indomethacin (MP Biomedicals, Santa Ana, CA, USA). The medium was refreshed every 3 days. The cells washed twice with PBS to remove medium and then fixed with 4% paraformaldehyde (Sigma-Aldrich, St. Louis, MO, USA). Then cells were washed gently three times with ddH2O and stained with Oil Red O (Sigma-Aldrich, St. Louis, MO, USA). Photographs were taken by the inverted optical microscope (Olympus, Tokyo, Japan).

## Establishment of experimental CIA model and treatment by infusion of MSCs

A total of 24 DBA/1 mice were randomly assigned to four experimental groups: collagen-induced arthritis (CIA) mice (CIA), CIA mice infused with UCs (UC), CIA mice infused with BMSCs (BMSC), and CIA mice infused with SHED (SHED). CIA model was induced as described previously (*Brand, Latham & Rosloniec, 2007*). Briefly, DBA/1 mice received an injection into the base of the tail with 100 μg of type II collagen (CII) (Chondrex, Redmond, WA, USA) emulsified in Freund's complete adjuvant. A total of 15 days later, they received booster injections of 100 μg of type II collagen in Freund's incomplete adjuvant. On day 30, $1 \times 10^6$ MSCs dissolved in 200 μl PBS per mouse were administered via tail vein injection to three groups (UC, BMSC, and SHED) and an equal volume of PBS was administered to the control group (CIA).

## Disease severity score

Each paw was evaluated for a period of every 3 days after the infusion of MSCs and were scored individually for the severity of arthritis using a previously described scoring system (*Chen et al., 2013*, *2018*). No evidence of erythema and swelling received a score of 0, erythema and mild swelling confined to the tarsals or metatarsals received a score of 1, erythema and moderate swelling of tarsal and the metatarsal or tarsal and ankle joints received a score of 2, erythema and severe swelling extending from the ankle to metatarsal joints received a score of 3, and erythema and severe swelling encompassing the ankle, foot, and digits, or ankylosis of the limb received a score of 4. Four paws scores added up to a total score per mouse.

## ELISA

Mice were sacrificed 30 days after MSC infusion. Mouse blood was collected at eye socket and serum was harvested after centrifugated at 3,000 rpm for 20 min. TNF-α and IL-1β levels in mice serum were measured using Enzyme-Linked Immunosorbent Assay (ELISA) kits (Yanhui Biotechnology, Shanghai, China) according to the manufacturer's instructions.

## Micro-CT analysis

Mice hind limbs were cut off after blood collection. Micro-computed tomographic (Micro-CT) scans of the hind limbs were performed using an explore Locus SP micro CT

scanner (GE, Boston, MA, USA) with the following scanning parameters: scanning resolution: 14 um; rotation angle: 360°; tube voltage: 80 kV; tube current, 80 μA; exposure time, 2,960 ms; everage frame: 4; and pixel combination: $1 \times 1$. The software program Micview V2.1.2 and ABA were used to create a 3D reconstruction and to analyze the bone. The measured area of bone volume was set with a length of one mm in the distal and proximal direction from the center of each metatarsophalangeal joint (*Chen et al., 2018*). The mean bone volume of the second to fourth metatarsal and phalangeal bones were compared.

## Histological analysis

Hind limbs were decalcified in 17% EDTA for 15 days after being fixed for 48 h in 4% paraformaldehyde at 4 °C. Then hind limbs were embedded in paraffin and serially sectioned into three μm thick paraffin sections. The sections were stained with haematoxylin–eosin (Leica Biosystems, Wetzlar, Germany) using standard protocols. Sections were observed under the light microscope (DM6B; Leica, Wetzlar, Germany). The extent of synovitis, pannus formation, and bone/cartilage destruction was determined using a graded scale, as follows: no signs of inflammation received a score of 0, mild inflammation with hyperplasia of the synovial lining without cartilage destruction received a score of 1, increasing degrees of inflammatory cell infiltration and cartilage/bone destruction received a score of 2–4 (*Chen et al., 2013*).

## Immunohistochemistry

The paraffin sections were incubated with primary antibodies TNF-$\alpha$ (1:200; Abcam, Cambridge, UK) and IL-1$\beta$ (1:200; Abcam, Cambridge, UK). Secondary antibodies (1:1000) were purchased from Vector Laboratories. The stained sections were observed using the light microscope (DM6B; Leica Microsystems, Heerbrugg, Switzerland). The photographs were evaluated by Image J (Media Cybernetics, Maryland, USA) from three randomly selected views of each specimen. Quantification of the number of positively stained cells or percentages of the positively stained area over the total area was performed using the Image J (National Institute of Health, Bethesda, MD, USA) software from three randomly selected views of each specimen.

## Statistical analysis

All numerical data are expressed as mean ± standard deviation of six mice per group from three independent experiments. Statistical analyses were performed using the Prism software (GraphPad, La Jolla, CA, USA). Statistical differences were assessed by one-way ANOVA for comparison among multiple groups and Tukey's multiple comparisons test for comparison between two groups. $P < 0.05$ was considered significant.

## RESULTS

### Isolation and characterization of MSCs

Purified MSCs were successfully obtained from bone marrow, umbilical cords, and dental pulps provided by donors. All the three MSC populations revealed positive expression (≥99% positive) of CD105, CD73, CD90, and negative expression (≤1% positive) of

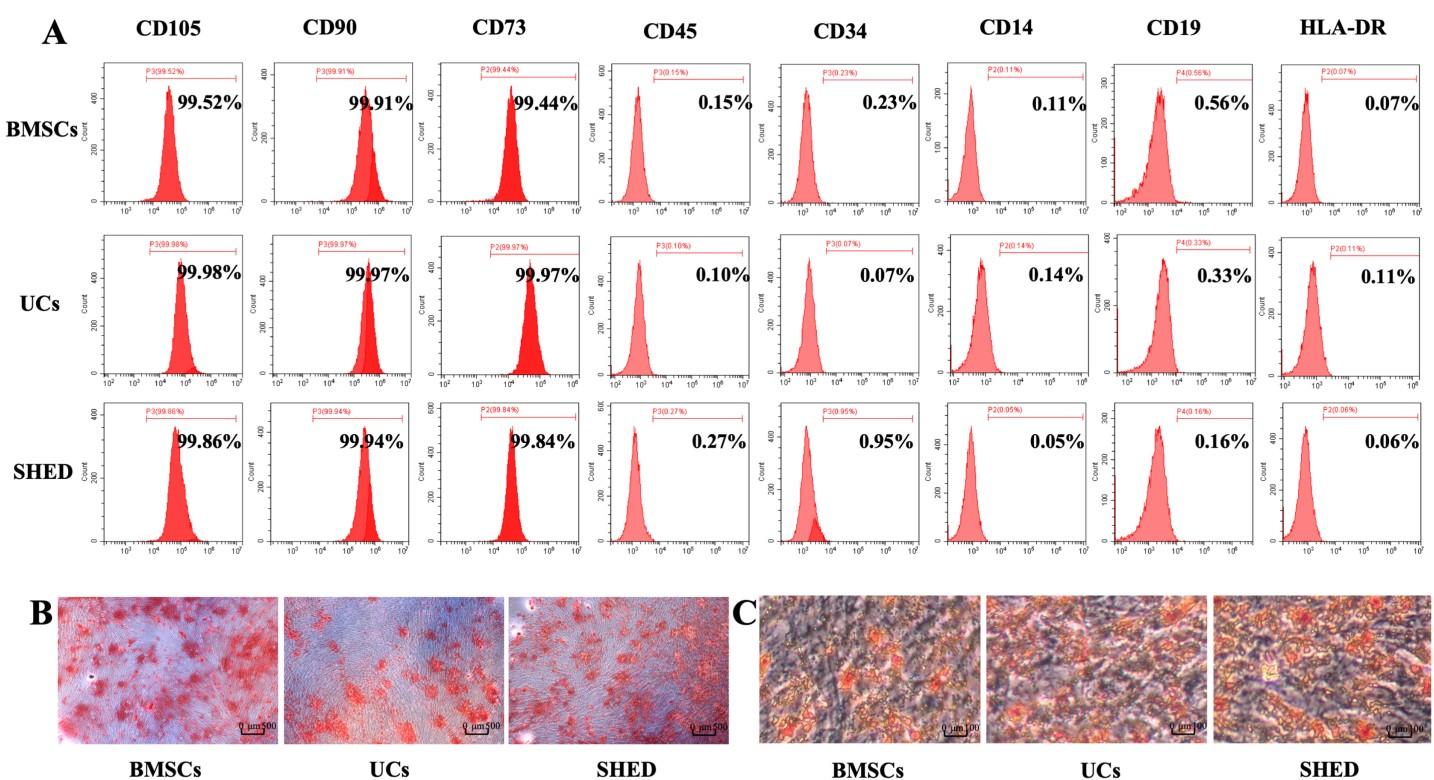

**Figure 1 Characterization of hUCMSCs, hBMSCs and hSHED.** (A) Flow cytometric analysis of ex-vivo-expanded hUCMSCs, hBMSCs, and hSHED. All the three MSC populations revealed positive expression of CD105, CD90, CD73, and negative expression of CD45, CD34, CD14, CD19, and HLA-DR. (B) After osteogenic induction for 28 days, hUCMSCs, hBMSCs, and hSHED formed mineralized nodules stained by Alizarin Red. (C) After culturing in adipogenesis inducing medium for 14 days, hUCMSCs, hBMSCs, and hSHED were found to form lipid droplets stained with Oil Red O.

CD45, CD34, CD14, CD19, and HLA-DR (Fig. 1A). Osteogenic and adipogenic differentiation assays were performed to investigate the differentiation potential of MSCs. After osteogenic induction for 28 days, mineralized nodules stained with Alizarin red were observed. (Fig. 1B). After culturing in adipogenesis inducing medium for 14 days, the MSCs were found to form lipid droplets by staining with Oil Red O (Fig. 1C). Based on the above results, the isolated MSCs met the ISCT criteria: adhering to plastic, expressing specific surface antigen and having multipotent differentiation potential (M et al., 2006).

## MSC treatment alleviate RA to different extent

We successfully established the CIA model on mice and treated the mice with $1 \times 10^6$ MSCs (Fig. 2A). The severity of RA was evaluated based on disease severity score, expression of inflammation cytokines, Micro-CT analysis and histological analysis. We recorded the severity score every 3 days after MSC treatment. The mice treated with UCs showed the most obvious decline in their severity score. However, the mice treated with BMSCs and SHED showed a lesser rate of decline in their severity score (Fig. 2B). The severity of RA was determined with Micro-CT analysis by measuring the shape and volume of bone (Fig. 3). Cartilage and bone destruction and inflammation in mice ankles were revealed in CIA mice. Specifically, the joint surfaces had a rough and irregular
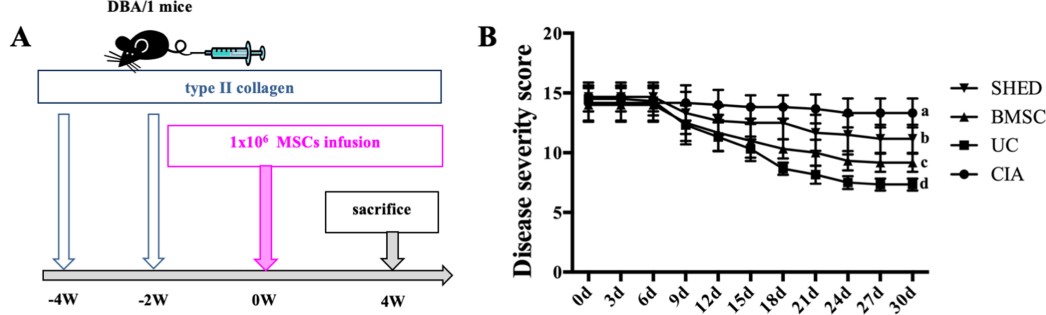

**Figure 2 MSC treatment alleviate RA to different extents.** (A) Schema indicating of the study design. Type II collagen was injected into mice twice via tail vein at −4 week and −2 week, respectively. $1 \times 10^6$ MSCs was infused into mice via tail vein at 0 week. All the mice were sacrificed at 4 week. (B) Disease severity score of four groups. The data indicate the mean ± SD of six mice per group from three independent experiments. The data were analyzed using two-way ANOVA for comparison among multiple groups and Tukey's multiple comparisons test for comparison between two groups ($P < 0.05$).

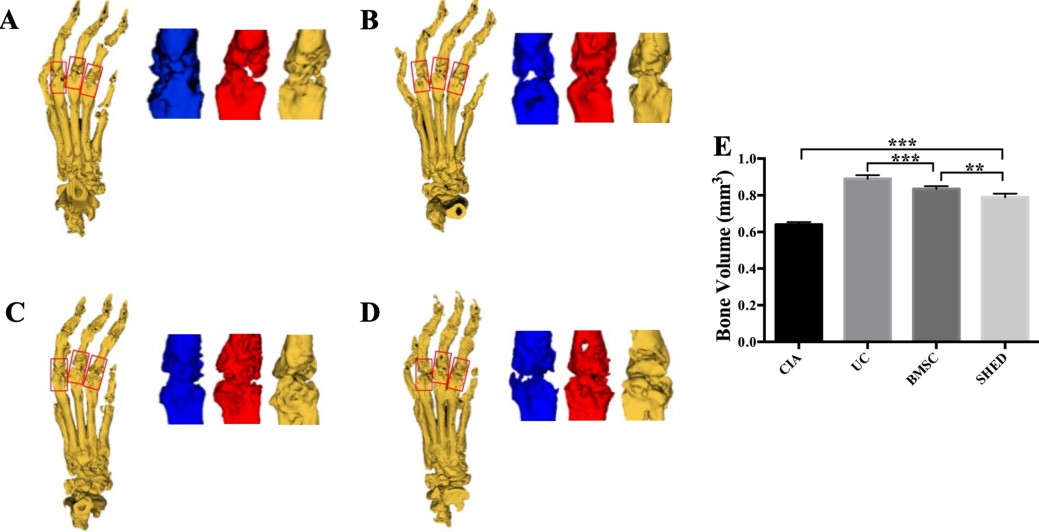

**Figure 3 MSC treatment prevented bone erosion in CIA mice.** The bone volume measured area were set with one mm length in the distal and the proximal direction from the center of each metatarso-phalangeal joint. The mean bone volume of second to fourth metatarsal and phalangeal bones were compared. (A) Bone erosion is observed in the CIA mice. (B) UC treatment reduced bone loss to the greatest degree. (C) BMSC treatment reduced bone loss to some extent. (D) SHED had the least treatment effect on reducing bone loss. (E) The mean bone volume of each group was evaluated and presented in column. The data indicate the mean ± SD of six mice per group from three independent experiments. The data were analyzed using one-way ANOVA for comparison among multiple groups and Tukey's multiple comparisons test for comparison between two groups (**$P < 0.01$, ***$P < 0.001$).

articular appearance with a diffuse lesion pattern on the articular surface (Fig. 3A). However, the number of lesions were lower in MSCs treated mice (Figs. 3B–3D). According to micro-CT image quantitative analyses, mice treated with UCs presented with the highest bone volume. Compared with those treated with BMSCs, the mice treated with SHED presented with a lower bone volume (Fig. 3E). Cartilage destruction and
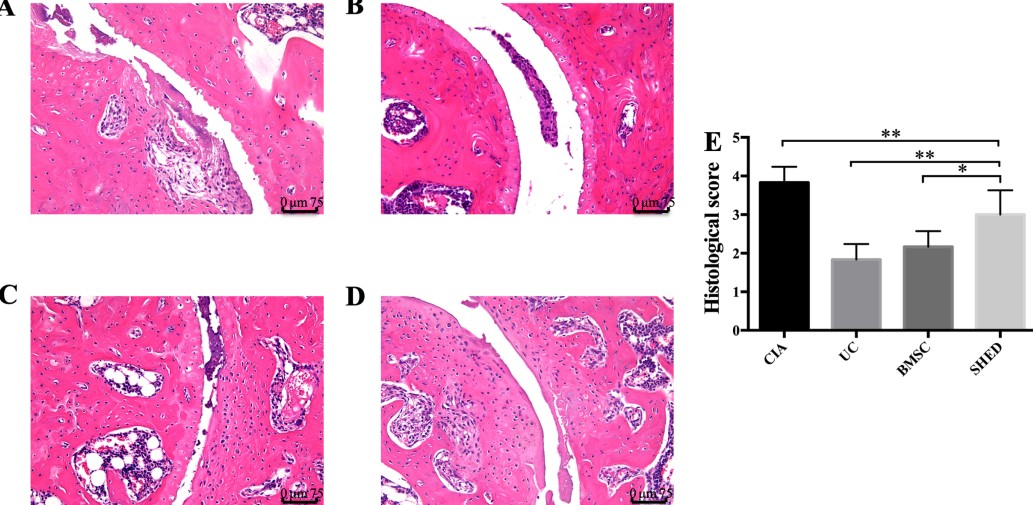

**Figure 4 Histopathological examination of hind ankle joints of mice.** (A) Joint destruction is observed in the CIA mice. (B) UC treatment reduced synovitis, pannus, erosion, and inflammatory cells infiltration to the greatest degree. (C) BMSC treatment reduced synovitis, pannus, erosion, and inflammatory cells infiltration to some extent. (D) SHED had the least treatment effect on reducing synovitis, pannus, erosion, and inflammatory cells infiltration. (E) The histological score of each group was evaluated and presented in column. Scale bar, 75 μm. The data indicate the mean ± SD of six mice per group from three independent experiments. The data were analyzed using one-way ANOVA for comparison among multiple groups and Tukey's multiple comparisons test for comparison between two groups (*$P < 0.05$, **$P < 0.01$). 

inflammation in mice ankles were determined by histopathological examination (Fig. 4). In CIA mice, the destruction of cartilage and bony structures were observed on the articular surfaces of the joint. In addition, synovial hyperplasia and inflammatory cell infiltration were observed on the articular surface of the joints from CIA mice (Fig. 4A). Mice treated with UCs showed less bone/cartilage destruction and inflammatory cell infiltration than the BMSCs and SHED treatment groups (Figs. 4B–4D). BMSCs and SHED infusion could also reduce bone/cartilage destruction and inflammatory cell infiltration in CIA mice, although BMSCs were more effective than SHED (Fig. 4E). According to previous studies, changes in the expression of the inflammatory cytokines were observed in the joints and serum of RA patients (*Fox, 2000*; *Lam et al., 2000*; *Noh et al., 2009*; *Darrieutort-Laffite et al., 2014*). This led to the investigation of the effect of MSC on the inhibition of the pro-inflammatory factors that are most closely related to RA. The three kinds of MSC treatments effectively inhibited the expression levels of TNF-α and IL-1β compared with the no treatment group (Fig. 5). UCs infusion reduced the TNF-α and IL-1β expression levels in joints and serum. Compared with UCs, the infusion of BMSCs showed a higher level in the expression of TNF-α and IL-1β. The expressions of TNF-α and IL-1β were higher after SHED infusion compared with the infusions of BMSCs and UCs (Fig. 5).

## DISCUSSION

According to previous studies, tail vein infusion, intraperitoneal infusion, and intra-articular implantation were three widely used methods to inject MSCs into mice. Tail vein infusion allows stem cells to circulate more quickly and efficiently with the blood to the

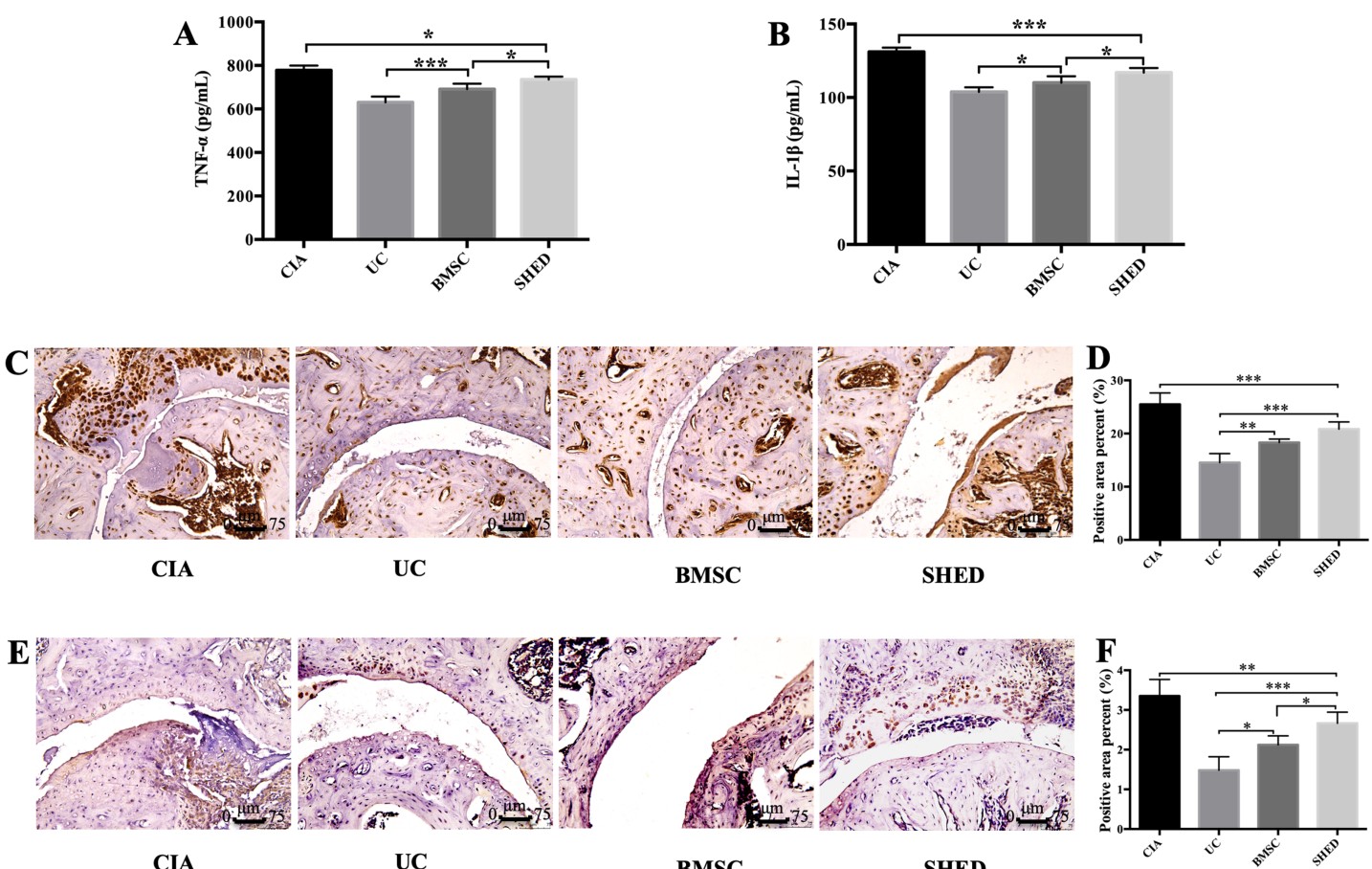

**Figure 5 MSCs reduced inflammatory cytokine expression in serum and hind limb ankle joins.** Level of serum total TNF-α (A) and IL-1β (B) was measured by ELISA. MSC treatment reduced expression of TNF-α and IL-1β. The expression of TNF-α and IL-1β decreased in UC group compared to BMSC group and SHED group, while the expression was lower in the BMSC group compared to the SHED group. Representative images of immunoreactivity for TNF-α (C and D) and IL-1β (E and F) in hind limb ankle joins of the mice. The expressions of TNF-α and IL-1β decreased in MSC treatment groups compared with CIA group. UC group showed lower TNF-α and IL-1β compared to the BMSC group or the SHED group, while the expression was lower in the BMSC group compared to the SHED group. The histological score of each group was evaluated and presented in column. Scale bar, 75 μm. The data indicate the mean ± SD of six mice per group from three independent experiments. The data were analyzed using one-way ANOVA for comparison among multiple groups and Tukey's multiple comparisons test for comparison between two groups (*$P < 0.05$, **$P < 0.01$, ***$P < 0.001$).

whole body, including the damaged tissue. Therefore, MSCs infused through tail vein could effectively reduce systemic inflammatory responses by paracrine effects, such as cell-cell interaction and cellular component transfer. Furthermore, tail vein infusion is more similar to the infusion method used in clinical treatment. Take all the above advantages into consideration, tail vein infusion was used in this study.

MSCs, lacking major histocompatibility II and several costimulatory molecules, are relatively immune-privileged, or at least are hypoimmunogenic, and have the ability to escape immune recognition. In the last several years, MSCs were shown to modulate the innate and adaptive immunity (*Glennie et al., 2005*; *Sato et al., 2007*). These cells may inhibit the function of the major immune cell populations, including dendritic cells, T cells, B cells, and natural killer cells. Meanwhile, the immunomodulatory and anti-inflammatory properties of MSCs have been tested in a variety of animal

models and have been applied in specific clinical settings (*Schüle & Berger, 2015*; *Al Jumah & Abumaree, 2012*). The ability of MSCs to quickly proliferate in vitro make it possible for MSCs to be applied in clinical treatments. Previous animal studies and clinical trials reported that MSCs may have a curative effect on some kinds of ADs (*Liang et al., 2016*).

The immunomodulatory ability of stem cells from different sources is controversial. UCs were considered to have better immunosuppressive effect on T cells, B cells, and NK cells than BMSCs (*Najar et al., 2010*; *Li et al., 2014*; *Yoo et al., 2009*; *Ribeiro et al., 2013*). Until now, there have been few reports on the therapeutic effects of SHED on ADs. SHED were reported to have similar characteristics for immune regulation to MSCs (*Alipour et al., 2013*; *Yamaza et al., 2010*). Furthermore, with advantages of outstanding proliferation capability, abundant cell supply, and painless stem cell collection, SHED were indicated to be a desirable option as a cell source for therapeutic applications (*Nakamura et al., 2009*). For these reasons above, the immunomodulatory functions of SHED were verified. In this study, we noticed the strong immunosuppressive ability of UCs in RA. The difference of immunosuppressive capabilities of different MSCs was consistent with the previous reports (*Najar et al., 2010*; *Li et al., 2014*; *Yoo et al., 2009*; *Ribeiro et al., 2013*; *Alipour et al., 2013*). The differences in immunomodulatory capacity may be due to that the three MSC populations come from different tissue (*Ribeiro et al., 2013*). The weaker systemic regulatory ability of SHED may be due to their development from the neural crest, which mainly differentiates into nerve cells or local tissue cells of the head.

In the course of RA, the dynamic balance of M1/M2 type macrophages was broken, leading to the increase of M1-type macrophage, which releases proinflammatory cytokines, such as TNF-α and IL-1β. MSCs could transform M1-type pro-inflammatory macrophage into M2-type anti-inflammatory macrophage through paracrine effects, thereby suppressing the inflammatory response (*Németh et al., 2009*). IL-1β and TNF-α have been widely recognized as the dominant proinflammatory cytokines of RA. TNF-α plays an important role in local synovial inflammation, pannus formation, and the tissue damage of RA (*Bradley, 2008*). A previous study suggests that TNF-α inhibitors can obviously reduce the disease activity in RA patients and improve the corresponding clinical symptoms (*Hyrich et al., 2004*). It has been recognized that high levels of IL-1β have been detected in the serum and joint fluid of RA patients. Its level was closely related to the activity and histopathological characteristics of the disease, such as synovial hyperplasia and leukocyte infiltration (*Noh et al., 2009*; *Darrieutort-Laffite et al., 2014*). To verify the effect of MSCs on the expression of inflammatory cytokines closely associated with CIA pathogenesis, TNF-α and IL-1β levels in joints and serum were determined. The levels of IL-1β and TNF-α were increased by CIA induction and decreased by treatment with MSCs. The decreases of these inflammatory markers in the joints were consistent with those in the serum. Thus, systemic injection of MSCs could relieve both systemic and joint inflammation. Our results indicate that systemic injection of UCs may be the most appropriate new treatment for ADs.

## CONCLUSIONS

In this study, we compared the therapeutic effects of BMSCs, UCs, and SHED on mice with CIA. Specifically, the therapeutic efficacy of UCs is better than BMSCs while that of BMSCs is better than SHED on RA, according to the extent of reducing bone resorption, joint destruction, and inflammatory factor expression. In studying the source of MSCs and their corresponding curative effect, UCs are a better choice than BMSCs and SHED in clinical treatment on ADs, or at least on RA.

### Funding

This work was supported by the grants from the National Key Research and Development Program of China (No.2016YFC1101400) and the National Natural Science Foundation of China (Nos., 81570976 and 81870768), the Scientific Young Alma of Shaanxi province (2018KJXX-015). The funders had no role in study design, data collection and analysis, decision to publish, or preparation of the manuscript.

### Grant Disclosures

The following grant information was disclosed by the authors:
National Key Research and Development Program of China: 2016YFC1101400.
National Natural Science Foundation of China: 81570976 and 81870768.
Scientific Young Alma of Shaanxi province: 2018KJXX-015.

### Competing Interests

The authors declare that they have no competing interests.

### Author Contributions

- Qing Zhang conceived and designed the experiments, performed the experiments, analyzed the data, contributed reagents/materials/analysis tools, prepared figures and/or tables, approved the final draft.
- Qihong Li performed the experiments, analyzed the data, prepared figures and/or tables, approved the final draft.
- Jun Zhu performed the experiments, approved the final draft.
- Hao Guo analyzed the data, contributed reagents/materials/analysis tools, approved the final draft.
- Qiming Zhai analyzed the data, contributed reagents/materials/analysis tools, approved the final draft.
- Bei Li authored or reviewed drafts of the paper, approved the final draft.
- Yan Jin conceived and designed the experiments, authored or reviewed drafts of the paper, approved the final draft.
- Xiaoning He conceived and designed the experiments, prepared figures and/or tables, authored or reviewed drafts of the paper, approved the final draft.
- Fang Jin conceived and designed the experiments, authored or reviewed drafts of the paper, approved the final draft.

## Animal Ethics

The following information was supplied relating to ethical approvals (i.e., approving body and any reference numbers):

All animal-handling procedures and experimental protocols performed were approved by the guidelines set forth by the Animal Care Committee of the Fourth Military Medical University, Xi'an, China (2018-kq-007).

## Data Availability

The raw data are available in the Supplementary Files.

## Supplemental Information

Supplemental information for this article can be found online at http://dx.doi.org/10.7717/peerj.7023#supplemental-information.

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
