# Peer review of "Comparison of therapeutic effects of different mesenchymal stem cells on rheumatoid arthritis in mice"

_PeerJ, doi:10.7717/peerj.7023_

## Round 0.1 · original submission · Major Revisions

While the reviewers overall looked positively at your manuscript they also raised some concerns which should be addressed by the authors. In particular a more detailed characterization of the MSC is required. Also there is some concern about the use of the English language and specific wording which could be improved in some sections of the manuscript. The reviewers also raised concern about the quality of some of the figures.

·

Basic reporting

The manuscript is well organized with clear scientific writing.

Experimental design

The research question and methods are well defined with hypothesis driven.

Validity of the findings

No comment

Additional comments

In this manuscript, the authors tried to compare different origins of MSCs in their immunomodulation ability, and used RA as an animal model to evaluate the treatment outcomes. Overall, the topic is important in the field, and the results are consistent through whole manuscript. I only have some suggestions in Discussion section.

1. As the authors mentioned that macrophage plays a major role in RA, the authors should discuss some potential mechanisms, in which how systemic MSC therapy inhibits macrophage activity?

2. In the first paragraph of Discussion, the authors claimed that systemic MSC infusion may be able to migrate into the damaged tissue. However, several recent publications have shown that paracrine effects, such as cell-cell interaction and cellular component transfer, likely be the major reasons in systemic MSC therapy. Please discuss.

3. The reason of using SHED in this study is not convincing. Craniofacial region-derived MSCs have unique advantages, such as easily accessible, outstanding proliferation ability, and so on. Please revise in the Discussion.

Reviewer 2 ·

Basic reporting

Article text is clear and easily understandable throughout. There are some minor errors as below
Abstract: Line 31: disease; Line 36: in mice; Line 38 and 47: MSC-treated
Introduction:
Line 55: Using “kind of chronic“ is unclear. Please address
Line 58:.. “articular or bone margin..
Line 65: “immune dysfunction”
Line 75: space after RA
Line 84: MSC
Line 88: MSC populations above…
Methods
Line 129: Use “1x106”
Results
Line 192 &199: MSC
Line 193: …the most obvious…
Discussion
Line 226: MSC
Line 238 and 240: …considered to have….reported to have…

Minor Issues:
Literature references, sufficient field background/context provided.
Introduction is written well and referenced appropriately for the most part. However, some studies that indicate that there are scenarios where the use of MSCs is not always positive in the context of RA, particularly when the cells are applied at a point where inflammation is high. Please provide references here.
Line 128: The authors should justify the use of the 6 surface markers use to characterise MSC populations (Line 128) and the selection made.
Line 142: Provide a reference for the scoring system used here rather than end of paragraph.
Line 166-167: “current standard being used” is not sufficient to describe a scoring system and is this scheme used by a lot of publications? Cite if so.

Professional article structure, figures, tables. Raw data shared.

Submission format is acceptable. Figures are relevant and images are of an acceptable quality and legible for the most part. The use of “remarkably” in some figure legends is inappropriate and should be changed. Raw data is shared as supplemental.
Please use more appropriate language – for example in figure 1 there are small although significant reductions in inflammatory cytokine levels and should be described as such with references to the statistical analysis performed. Please also indicate the number of mice used (n=?) in all figures where relevant.

Minor Comments
Figure 2: Indicate what collagen type used; Use “MSC treatment …to different extents”. Please indicate the number of replicates used where relevant in all figures as well
Figure 3: Please label the graphs representing histological scores as B and D and address the legend appropriately.
Rephrase “remarkable”.
“… cytokine expression in hind limb ankle joints.”
“… UC treatment..”
Y-axis legends are not easily readable and font should be increased.
Figure 4 and 5:
“MSC treatment…… . UC treatment….. BMSC treatment”
(D). “SHED had….”.
Re-phrase “remarkable”.

Experimental design

The research described has some novelty with respect to its comparative nature but is iterative.
The research question is defined and has some relevance to the field in the context of the cell comparison performed. This particular comparison has not been performed previously to my knowledge.
Ethical approvals for pre-clinical experiments was sought with a reference to same in Methods. Overall, methods used were well described.

Validity of the findings

Submission format is acceptable. Figures are relevant and images are of an acceptable quality and legible for the most part. The use of “remarkably” in some figure legends is inappropriate and should be changed. Raw data is shared as supplemental.
Please use more appropriate language – for example in figure 1 there are small although significant reductions in inflammatory cytokine levels and should be described as such with references to the statistical analysis performed. Please also indicate the number of mice used (n=6) in all figures where relevant.

Minor Comments
Figure 2: Indicate what collagen type used; Use “MSC treatment …to different extents”. Please indicate the number of replicates used where relevant in all figures as well
Figure 3: Please label the graphs representing histological scores as B and D and address the legend appropriately.
Rephrase “remarkable”.
“… cytokine expression in hind limb ankle joints.”
“… UC treatment..”
Y-axis legends are not easily readable and font should be increased.
Figure 4 and 5:
“MSC treatment…… . UC treatment….. BMSC treatment”
(D). “SHED had….”.
Re-phrase “remarkable”.

Additional comments

Overall, the results found were valid with statistical significance and some novelty exists. The number of animals (n=6) did seem low for the model but significant data was generated.
Data is robust and statistics are handled well.

Conclusions are well stated, linked to original research question & limited to supporting results. The use of remarkable should be deleted here as well though (line 260) – a clear over statement of data. A statistical reference should be sufficient throughout.

·

Basic reporting

The authors are presenting a work that aims at clarifying that several sources of MSC provide different outcomes regarding RA treatment. This subject is clinically relevant and with great impact in helping the decision on the MSCs selection for future clinical use. However, the approach adopted was not innovative, nor presents new methodologies, or possible mechanism that could increase the knowledge in the field.
Nevertheless, given the relevance of the work some crucial improvements are suggested.

Overall, the English language should be improved to ensure that an international audience can clearly understand your text. For example, the authors should avoid terms such as “is a kind” (line 56).
The structure of the article is under an acceptable format of ‘standard sections’ and the figures are relevant to the content of the article. Nevertheless, the resolution of the graph Fig 2B needs to be improved in order to distinguish the labeling and the figure legends could be improved.

Experimental design

The original research is within the aims and scope of the journal; and the research question is relevant. However, it has been briefly support by the current knowledge and the some technical issues have been identified, as follows:

INTRODUCTION

1. The introduction needs more detail as well. I suggest that you improve the description at lines 78-85 to provide more justification for using MSC as a therapeutic strategy for RA. Specifically, you should detail the mechanisms by which MSC can indeed be a better alternative regarding the current treatment options. Moreover, the clinical aspects of RA should also be detailed and clearly articulated with the potential effect of the MSCs.

2. SHED cells are derived from adult tissues therefore it is not clear what the authors meant by “SHED are obtained from relatively primitive tissues” and “with a greater potential for differentiation” (lines 94-95). Please comment.


MATERIALS and METHODS

3. The materials and methods section should be greatly improved and detailed. Specifically:

4- The ethical issues regarding the MSCs collection are missing.

5- The isolation and characterization of the MSCs must be more detailed. The methodologies adopted have not been justified nor supported with literature.

6- The establishment of experimental CIA model has only been briefly described in the manuscript (lines 136-137). This information should be included in the materials and methods section in detail.

7- The authors should justify the MSCs administration conditions. Specifically, the authors should justify the quantity of cells that were administered, the timing (why day 30?), and the number of times the cells were administered (only once?).

8- The timing and collection of mice serum for measuring the levels of TNF-α and IL-1β are missing. The same applies for the joint samples, that are not mentioned in the M&M section and are described only for the first time in the results section (line 197).


RESULTS SECTION

The first subsection from the results (“Flow cytometry evaluation of surface markers of MSCs”) is very limited. The authors should perform a more in deep analyses to confirm that the isolated cells are indeed MSCs. Namely, that the cells fulfill the ISCT criteria, ie, show three lineage differentiation capacity and are positive for MSC specific markers. Otherwise, with the current available data it is not clear that the isolated cells are MSCs. Importantly, the purity of the isolated populations is also not shown.

The subsections “MSCs treatment reduced disease severity score in RA”, “MSCs treatment reduced inflammatory responses in CIA”, “MSCs treatment prevented tissue damage in CIA”, should be merged and the results articulated.


DISCUSSION SECTION

Some sentences are not supported by literature data, nor from results obtained by the authors (e.g. The sentence “UCs have the strongest immunosuppressive ability, possibly because they are more primitive cells compared to BMSCs and SHED.” (Lines 245-246)).

Validity of the findings

9. Overall, the methodology adopted, and results obtained are briefly addressed or missing, which limits the interpretation of the data. For example, the purity of the cell population obtained after the isolation process was not evaluated. This would be important for relating the effects observed with the administered cells. Moreover, the MSCs were administered i.v. therefore the authors should monitor the cells within the organism (to decipher the mechanism of action of the cells it is important to follow cells survival during the experiment time, if the cells migrate to the lesions site, or if cells are kept in other tissues (e.g. the lung)).

---

## Round 0.2 · accepted · Accept

The authors have significantly improved the quality of the manuscript following the advice of the reviewers

# ·

Basic reporting

Overall, the authors have addressed all the reviewr's request and concerns. There has been a good effort ot improve the manuscript that has now a clear and appropriate scientific language. The introduction section, M&M, results and discussion have also been greatly improved and detailed. Therefore, I believe the manuscript is acceptable for publication.

Experimental design

Please see the comment above.

Validity of the findings

Please see the comment above.

Additional comments

Please see the comment above.